# Effects of Dementia on Outcomes after Cervical Spine Injuries in Elderly Patients: Evaluation of 1512 Cases in a Nationwide Multicenter Study in Japan

**DOI:** 10.3390/jcm12051867

**Published:** 2023-02-27

**Authors:** Yohei Yamada, Noriaki Yokogawa, Satoshi Kato, Takeshi Sasagawa, Hiroyuki Tsuchiya, Hiroaki Nakashima, Naoki Segi, Sadayuki Ito, Toru Funayama, Fumihiko Eto, Akihiro Yamaji, Junichi Yamane, Satoshi Nori, Takeo Furuya, Atsushi Yunde, Hideaki Nakajima, Tomohiro Yamada, Tomohiko Hasegawa, Yoshinori Terashima, Ryosuke Hirota, Hidenori Suzuki, Yasuaki Imajo, Shota Ikegami, Masashi Uehara, Hitoshi Tonomura, Munehiro Sakata, Ko Hashimoto, Yoshito Onoda, Kenichi Kawaguchi, Yohei Haruta, Nobuyuki Suzuki, Kenji Kato, Hiroshi Uei, Hirokatsu Sawada, Kazuo Nakanishi, Kosuke Misaki, Hidetomi Terai, Koji Tamai, Akiyoshi Kuroda, Gen Inoue, Kenichiro Kakutani, Yuji Kakiuchi, Katsuhito Kiyasu, Hiroyuki Tominaga, Hiroto Tokumoto, Yoichi Iizuka, Eiji Takasawa, Koji Akeda, Norihiko Takegami, Haruki Funao, Yasushi Oshima, Takashi Kaito, Daisuke Sakai, Toshitaka Yoshii, Tetsuro Ohba, Bungo Otsuki, Shoji Seki, Masashi Miyazaki, Masayuki Ishihara, Seiji Okada, Shiro Imagama, Kota Watanabe

**Affiliations:** 1Department of Orthopedic Surgery, Graduate School of Medical Sciences, Kanazawa University, Ishikawa 920-8641, Japan; 2Department of Orthopedic Surgery, Toyama Prefectural Central Hospital, Toyama 930-8550, Japan; 3Department of Orthopedic Surgery, Nagoya University, Graduate School of Medicine, Nagoya 466-8550, Japan; 4Department of Orthopedic Surgery, Faculty of Medicine, University of Tsukuba, Ibaraki 305-8575, Japan; 5Department of Orthopedic Surgery, Graduate School of Comprehensive Human Sciences, University of Tsukuba, Ibaraki 305-8575, Japan; 6Department of Orthopedic Surgery, Ibaraki Seinan Medical Center Hospital, Ibaraki 306-0433, Japan; 7Department of Orthopedic Surgery, National Hospital Organization Murayama Medical Center, Tokyo 208-0011, Japan; 8Department of Orthopedic Surgery, Keio University School of Medicine, 35 Shinanomachi, Tokyo 160-8582, Japan; 9Department of Orthopedic Surgery, Graduate School of Medicine, Chiba University, Chiba 260-8670, Japan; 10Department of Orthopedics and Rehabilitation Medicine, Faculty of Medical Sciences University of Fukui, Fukui 910-1193, Japan; 11Department of Orthopedic Surgery, Hamamatsu University School of Medicine, Shizuoka 431-3192, Japan; 12Department of Orthopedic Surgery, Nagoya Kyoritsu Hospital, Aichi 454-0933, Japan; 13Department of Orthopedic Surgery, Sapporo Medical University, Sapporo 060-8543, Japan; 14Department of Orthopedic Surgery, Matsuda Orthopedic Memorial Hospital, Sapporo 001-0018, Japan; 15Department of Orthopedic Surgery, Yamaguchi University Graduate School of Medicine, Yamaguchi 755-8505, Japan; 16Department of Orthopedic Surgery, Shinshu University School of Medicine, Nagano 390-8621, Japan; 17Department of Orthopedics, Graduate School of Medical Science, Kyoto Prefectural University of Medicine, Kawaramachi-Hirokoji, Kyoto 602-8566, Japan; 18Department of Orthopedics, Saiseikai Shiga Hospital, Shiga 520-3046, Japan; 19Department of Orthopedic Surgery, Tohoku University Graduate School of Medicine, Miyagi 980-8574, Japan; 20Department of Orthopedic Surgery, Graduate School of Medical Sciences, Kyushu University, Fukuoka 812-8582, Japan; 21Department of Orthopedic Surgery, Nagoya City University Graduate School of Medical Sciences, Nagoya 467-8601, Japan; 22Department of Orthopedic Surgery, Nihon University Hospital, Tokyo 101-8393, Japan; 23Department of Orthopedic Surgery, Nihon University School of Medicine, Tokyo 173-8610, Japan; 24Department of Orthopedics, Traumatology and Spine Surgery, Kawasaki Medical School, Okayama 701-0192, Japan; 25Department of Orthopedic Surgery, Osaka Metropolitan University Graduate School of Medicine, Osaka 545-8585, Japan; 26Department of Orthopedic Surgery, Kitasato University School of Medicine, 1-15-1, Kanagawa 252-0374, Japan; 27Department of Orthopedic Surgery, Kobe University Graduate School of Medicine, Kobe 650-0017, Japan; 28Department of Orthopedic Surgery, Kochi Medical School, Kochi University, Nankoku 783-8505, Japan; 29Department of Orthopedic Surgery, Graduate School of Medical and Dental Sciences, Kagoshima University, Kagoshima 890-8520, Japan; 30Department of Orthopedic Surgery, Gunma University Graduate School of Medicine, 3-39-22 Showa, Maebashi 371-8511, Japan; 31Department of Orthopedic Surgery, Mie University Graduate School of Medicine, Mie 514-8507, Japan; 32Department of Orthopedic Surgery, School of Medicine, International University of Health and Welfare, Chiba 286-0124, Japan; 33Department of Orthopedic Surgery, International University of Health and Welfare Narita Hospital, Chiba 286-0124, Japan; 34Department of Orthopedic Surgery and Spine and Spinal Cord Center, International University of Health and Welfare Mita Hospital, Tokyo 108-8329, Japan; 35Department of Orthopedic Surgery, The University of Tokyo Hospital, Tokyo 113-8655, Japan; 36Department of Orthopedic Surgery, Osaka University Graduate School of Medicine, Osaka 565-0871, Japan; 37Department of Orthopedics Surgery, Surgical Science, Tokai University School of Medicine, Kanagawa 259-1193, Japan; 38Department of Orthopedic Surgery, Tokyo Medical and Dental University, Yushima 113-8519, Japan; 39Department of Orthopedic Surgery, University of Yamanashi, Yamanashi 409-3898, Japan; 40Department of Orthopedic Surgery, Graduate School of Medicine, Kyoto University, Kyoto 606-8501, Japan; 41Department of Orthopedic Surgery, Faculty of Medicine, University of Toyama, Toyama 930-0194, Japan; 42Department of Orthopedic Surgery, Faculty of Medicine, Oita University, Oita 879-5593, Japan; 43Department of Orthopedic Surgery, Kansai Medical University Hospital, Osaka 573-1191, Japan

**Keywords:** cervical spine injury, older adults, dementia, short- to middle-term outcome, functional prognosis, complications, mortality, propensity score matching

## Abstract

We aimed to retrospectively investigate the demographic characteristics and short-term outcomes of traumatic cervical spine injuries in patients with dementia. We enrolled 1512 patients aged ≥ 65 years with traumatic cervical injuries registered in a multicenter study database. Patients were divided into two groups according to the presence of dementia, and 95 patients (6.3%) had dementia. Univariate analysis revealed that the dementia group comprised patients who were older and predominantly female and had lower body mass index, higher modified 5-item frailty index (mFI-5), lower pre-injury activities of daily living (ADLs), and a larger number of comorbidities than patients without dementia. Furthermore, 61 patient pairs were selected through propensity score matching with adjustments for age, sex, pre-injury ADLs, American Spinal Injury Association Impairment Scale score at the time of injury, and the administration of surgical treatment. In the univariate analysis of the matched groups, patients with dementia had significantly lower ADLs at 6 months and a higher incidence of dysphagia up to 6 months than patients without dementia. Kaplan–Meier analysis revealed that patients with dementia had a higher mortality than those without dementia until the last follow-up. Dementia was associated with poor ADLs and higher mortality rates after traumatic cervical spine injuries in elderly patients.

## 1. Introduction

The rate of cervical spine injuries continues to rise in developed countries as the population ages [1]. This increase is primarily because the epidemiology of cervical spine injuries has shifted from younger patients to older ones, with ground-level falls becoming the primary mechanism of injury [1]. Cervical spine injuries are considered one of the most severe types of injury because of the risk of traumatic spinal cord injury, permanent neurological disability, and death [2,3,4,5,6]. The financial burden on patients, their families, and health care systems is enormous [7,8]. The lack of knowledge on cervical spine injuries in the elderly and of countermeasures against these injuries in this patient group is a global problem that needs to be addressed.

On the other hand, the prevalence of dementia is increasing in an aging society. It was estimated that 35.6 million people were living with dementia worldwide in 2010; this number is expected to almost double every 20 years and to reach 65.7 million in 2030 and 115.4 million in 2050 [9,10]. Existing research recognizes the increased risk of falls and fractures in patients with dementia, which is attributed to impaired motor function and balance [11]. Patients with dementia face difficulties in recovering their previous physical independence after sustaining fractures compared with those without dementia [12]. Moreover, dementia shares several common risk factors with osteoporosis, contributing to the high prevalence of osteoporosis and an increased risk of fractures [13,14]. Additionally, patients with dementia have a higher incidence of hip fractures, and on sustaining such fractures, these patients have a higher postoperative mortality rate and greater potential for limitations in activities of daily living (ADLs) than patients without dementia [15,16,17,18].

Although there have been numerous reports of dementia as a risk factor for poor prognosis after traumatic fractures, no study has reported on the effects of dementia on cervical spine injury outcomes. Therefore, in this study, we aimed to investigate the clinical characteristics of the cervical spine injuries in patients with dementia and the effects of dementia on functional and life outcomes following these injuries.

## 2. Materials and Methods

### 2.1. Study Participants

We retrospectively analyzed data from elderly patients (≥65 years at the time of injury) who had suffered a traumatic cervical spine injury between 2010 and 2020 who were enrolled in a multicenter database maintained by the Japan Association of Spine Surgeons with Ambition (JASA) [19]. Cervical spine injuries included cervical spine fractures, dislocations, spinal cord injuries, and combinations of these injuries. The criteria for inclusion in the database were patients who required in-hospital treatment and those who were followed up for at least 3 months.

### 2.2. Variables, Classifications, and Outcomes

Variables included patient demographics (age, sex, body mass index [BMI], residence status, pre-injury ADLs, pre-injury medical comorbidities apart from dementia, and the modified 5-item frailty index [mFI-5]), the results of blood tests performed on admission (total protein and hemoglobin [Hb]), injury status (presence of cervical fracture/dislocation, neurological impairment, and associated injury), and treatment status (steroid administration and surgery). The 5-mFI is a surrogate index of frailty consisting of five items: four comorbidities and one ADL indicator. The four comorbidities are hypertension, diabetes mellitus, congestive heart failure, and chronic obstructive pulmonary disease, while the ADL indicator is the status of needing assistance in daily living [20,21]. Neurological impairment was assessed using the American Spinal Injury Association Impairment Scale (AIS) [22]. External causes of injury were classified as falls at the ground level and others. The outcomes of interest were mortality, complication rate, and ADLs at 6 months post-injury. ADLs were evaluated based on the ability to walk independently, and independent walking was defined as walking with or without a cane. The dementia group included patients with a history of dementia, which was identified by retrospectively evaluating their medical records.

### 2.3. Statistical Analysis

Patients were divided into two groups according to the presence of dementia. Quantitative data were presented as the mean ± standard deviation, and the Shapiro–Wilk test was used to assess the normality of the data distribution. The differences in continuous variables between the groups were examined using the Student’s *t*-test for the parametric data and the Mann–Whitney U test for nonparametric data, while categorical data were compared using the chi-square test.

Propensity score matching analyses were performed to compare the complication rates and ADLs at 6 months post-injury between the patients with and without dementia. Propensity score matching analysis is widely used in retrospective cohort studies to control for confounding biases [23]. In the present study, the propensity scores were calculated based on five variables: age, sex, pre-injury ADLs (independent walking or non-independent walking), AIS score at the time of injury (AIS A or B, AIS C or D, or no spinal cord injury), and the implementation of surgical treatment. Patients with missing data at 6 months post-injury were excluded from the propensity score matching analysis. Furthermore, mortality up to the last follow-up was investigated using the Kaplan–Meier method. IBM SPSS Statistics for Macintosh, Version 27.0 (IBM Corp, Armonk, NY, USA), was used for all statistical analyses. The level of statistical significance was set at *p* < 0.05.

## 3. Results

### 3.1. Patient Characteristics Including Injury and Treatment Status

A total of 1512 patients (1007 males and 505 females; mean age, 75.8 ± 6.9 years; mean follow-up period, 19.1 ± 21 months) were included and evaluated in this study. Of these patients, 95 (6.3%) had dementia. Patients in the dementia group were older and predominantly female compared to those in the non-dementia group (Table 1). Patients with dementia had lower BMI, lower rates of living at home, lower pre-injury ADLs, a larger number of comorbidities (apart from Parkinson’s disease and diabetes mellitus), a higher mFI-5, and lower Hb levels than patients without dementia; they also used a larger number of medications than patients without dementia.

Regarding injury and treatment status, there was a larger proportion of patients who had sustained injuries as a result of ground-level falls and of those who had cervical spine fractures in the dementia group than in the non-dementia group; furthermore, fewer patients in the dementia group received surgical treatment than those in the non-dementia group (Table 2). There were no significant differences in the presence and severity of neurological impairment, dislocation, and associated injuries and in the administration of steroids between the groups.

### 3.2. Propensity Score Matching Analysis of Complications, ADLs, and Survival after Injury

Sixty-one pairs of patients were selected through propensity score matching. There were no significant differences in the five variables used for propensity score matching between the groups (Table 3). Propensity score matching analysis revealed that a significantly larger proportion of patients had dysphagia and lower ADLs at 6 months post-injury in the dementia group than in the non-dementia group (Table 4). Although there were no significant differences in the other complications at 6 months post-injury between the groups, the Kaplan–Meier method revealed that patients in the dementia group had a significantly poorer life expectancy than those in the non-dementia group (Figure 1).

## 4. Discussion

In this study, 1512 patients with a cervical spine injury aged ≥65 years, who were registered in a multicenter study conducted in Japan, were retrospectively examined; we aimed to evaluate the characteristics of patients with dementia who sustained cervical spine injuries and the effects of dementia on functional and life expectancy. To the best of our knowledge, this is the first study to examine the clinical characteristics of traumatic cervical spine injury in elderly patients according to the presence of dementia.

### 4.1. Characteristics of Cervical Spine Injury in Elderly Patients with Dementia

Patients in the dementia group tended to be older and to have poorer pre-injury health (e.g., medical comorbidities and frailty) than those in the non-dementia group. There were a larger number of elderly females in the dementia group than in the non-dementia group; this may be because among the elderly population, the prevalence of dementia is generally higher in females than in males, and this prevalence increases exponentially with age [24,25]. The reason for the larger number of comorbidities observed in the dementia group is that dementia is primarily a geriatric disease and is often comorbid with other geriatric diseases [26,27,28,29,30]. There is evidence to support an association between dementia and cardiovascular risk factors such as hypertension and dyslipidemia [30,31,32]. Furthermore, dementia is characterized by a decline in cognitive and physical function, which worsens as the disease progresses and is associated with a decrease in the ability to function independently [33,34,35,36]. Patients with dementia have a slower gait, poorer balance, an increased risk of falls, and decreased ADLs [37,38,39]. Therefore, in this study, the dementia group tended to have lower pre-injury ADLs than the non-dementia group.

In terms of injury characteristics, a larger number of injuries in the dementia group were caused by minor external forces such as ground-level falls than those in the non-dementia group. Consequently, patients with dementia sustained a larger number of cervical spine fractures but had fewer cervical spinal cord injuries and underwent fewer surgical interventions. This may be due to the fact that these patients had a higher rate of osteoporosis [40,41]. Yokokawa et al. examined the differences in the clinical outcomes of cervical spine injuries caused by external factors and reported that the patient group with injuries caused by lower external forces such as ground-level falls had more severe frailty and a worse prognosis than other patient groups [19]. These findings are consistent with the results of this study.

### 4.2. Effects of Dementia on Clinical Outcomes Following a Cervical Spine Injury

As mentioned previously, patients with dementia were older and in a poorer condition than patients without dementia; this made a direct comparison of outcomes difficult. Therefore, we adjusted for age, sex, pre-injury walking capacity, status of cervical cord injury, and the administration of surgical treatment and compared the matched groups using a propensity score matching analysis. The results showed that there was a significantly higher incidence of dysphagia and a lower rate of independent walking, as assessed 6 months post-injury, in the dementia group than in the non-dementia group. It has been reported that the incidence of aspiration after cervical spine injury is as high as 30%, and the risk factors are considered to be severe paralysis, tracheostomy, advanced age, and laryngeal edema [42]. Dysphagia has also been reported to occur more frequently in patients with dementia, with an incidence ranging from 13 to 57% [43,44]. Dysphagia in patients with dementia may be caused by decreased sensory and motor function, but the etiology has not been clarified. In a report of a patient with dementia who underwent a swallow angiogram, delayed pharyngeal and laryngeal responses or reflexes and prolonged intraoral phases were observed, and the severity of dysphagia was correlated with the severity of dementia [44]. Dysphagia is recognized as one of the major pathophysiologies that cause pneumonia, and pneumonia is one of the leading causes of morbidity and mortality from infectious diseases in the elderly [45,46]. Although not shown in this study, the most common cause of death in this patient database was pneumonia. We believe that patients with cervical spine injuries who have dementia should be provided with appropriate swallowing training, bearing in mind that dysphagia might occur [47]. Although there were no significant differences in the presence of complications apart from dysphagia between the groups, the rates of several of these complications tended to be higher in patients with dementia, suggesting that these patients may be at a higher risk of several complications. Regarding ADLs, despite the slightly smaller proportion of patients with neurological impairment in the dementia group, the level of walking independence at 6 months was significantly lower in this group than in the non-dementia group. This suggests that post-traumatic recovery is slower in patients with dementia and that a more intensive and specialized rehabilitation strategy needs to be established [12]. There have been reports of the effectiveness of enhanced rehabilitation programs in hip fracture patients with dementia [48], and there is a need to accumulate more knowledge about effective rehabilitation in such patients with cervical spinal injuries.

Life expectancy was significantly poorer in the dementia group after adjusting for background factors. In general, cervical spine injuries in the elderly have a poor prognosis, with high rates of mortality and complications [49]. Dementia is also known to have a poor prognosis. The risk of death in patients with dementia is reported to be approximately 2.6–2.9 times higher than that in patients without dementia [50,51]. Five-year survival rates in patients with dementia have been reported to be similar to or lower than those in patients with acute ischemic heart disease or cancer in most age groups [50,52]. Furthermore, there are many studies on the relationship between dementia and fractures in the case of hip fractures. Dementia has been identified as a factor that increases the risk of sustaining hip fractures, delays functional recovery, and worsens the 1- and 2-year life expectancy [15,16,17,18]. Conversely, traumatic injuries such as fractures and spinal cord injuries have been reported to be involved in the onset and exacerbation of dementia [53,54]. Thus, cervical spine injury and dementia are considered to have a mutually adverse relationship. Therefore, in cases of cervical spine injuries, even in the absence of cervical spinal cord injury or obvious neurological damage, the trauma may cause further deterioration of cognitive function, thereby adversely affecting life and functional prognosis.

### 4.3. Fall Trauma Risk in Elderly Patients with Dementia

Elderly women aged ≥75 years with dementia fall 2–3 times more often than cognitively healthy elderly people [55], and 70–80% of elderly patients aged ≥ 65 years with dementia fall annually [56]. A cohort study of 793 patients aged ≥ 75 years with dementia reported that 303 patients suffered fall trauma (mean follow-up, 3.7 years). A total of 328 fall trauma injuries occurred, of which 3.4% (11/328) were reported as intracranial injuries, 7.6% (25/328) as vertebral fractures, and 37.2% (122/328) as lower extremity fractures [57]. However, we were unable to find any report on the incidence of cervical spine injury in patients with dementia. Therefore, this is a subject for future study. In this study, the prevalence of dementia in patients with cervical spine injuries was 6.3%. Previous studies have reported a prevalence of dementia in elderly patients aged ≥ 65 years with cervical spine fractures ranging from 5–13%, which was consistent with the results of our study [58,59].

### 4.4. Prevention of and Enhanced Care for Cervical Spine Injury in Elderly Patients with Dementia

Our findings indicate that patients with dementia and cervical spine injury are frailer at the time of injury than patients with cervical spine injury without dementia; furthermore, these patients have worse functional and life prognoses and require more attention during initial treatment and prognostic management. The establishment of fall prevention programs and multidisciplinary treatment strategies for patients with dementia is required to prevent cervical spine injuries. Effective reinforcement or rehabilitation of cervical spine injuries in patients with dementia is warranted after the injury. In particular, complications such as dysphagia may contribute to mortality, and a swallowing assessment and appropriate swallowing training are needed.

### 4.5. Limitations

The strength of this study is that it examined the characteristics and clinical outcomes of cervical spine injuries in elderly patients with and without dementia in a nationwide multicenter study. However, the study has several limitations including the retrospective design and the fact that this was not an exhaustive study. Second, the diagnostic criteria, severity, etiology, and treatment of dementia were not investigated. In addition, this study was conducted in Japan, a country with the unique background of having the world’s largest aging population. However, the world is moving toward a hyper-aged society, and cervical spine injuries in the elderly are an urgent issue to be addressed. We believe that this study presents a model case of an aging society, and the results provide valuable insights that could aid in formulating preventive strategies against these injuries in the elderly.

## 5. Conclusions

This study retrospectively examined elderly patients with cervical spine injuries. We found that the patients in the dementia group were older and predominantly female and had lower BMI, lower walking capacity, and a higher 5-mFI than those in the non-dementia group. In other words, the frailty of patients in the dementia group was more severe. After adjusting for age, sex, pre-injury ADLs, AIS score at the time of injury, and administration of surgical treatment, we found that patients in the dementia group had a lower ability to walk 6 months after injury, a higher incidence of dysphagia, and worse short- to middle-term life expectancy than those in the non-dementia group. Dementia significantly affected the clinical outcomes in elderly patients with cervical spine injuries.

## Figures and Tables

**Figure 1 jcm-12-01867-f001:**
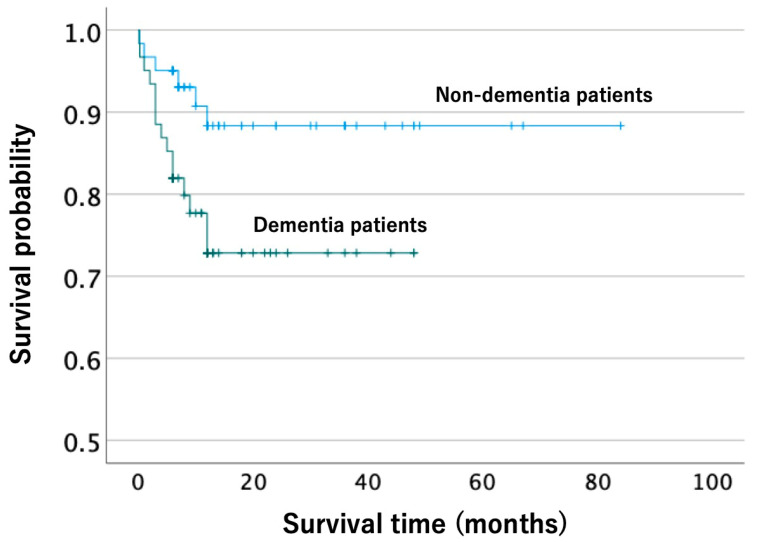
Kaplan–Meier survival curve showing the survival probability of patients with and without dementia. The mean survival time was 36.4 (95% CI: 31.4–41.5) months in the dementia group and 74.9 (95% CI: 68.0–81.8) months in the non-dementia group. The log-rank test revealed that patients with dementia had a lower survival rate than those without dementia (log-rank test: *p* = 0.031). CI, confidence interval.

**Table 1 jcm-12-01867-t001:** Comparison of the patient demographics between the dementia and non-dementia groups.

	Dementia (*n* = 95)	Non-Dementia (*n* = 1417)	*p*-Value
**Age at injury (years), mean ± SD**	82.1 ± 6.8	75.4 ± 6.7	<0.001 *
**Sex: Female, *n* (%)**	45 (47.4)	460 (32.5)	0.003 *
**Body mass index (kg/m^2^), mean ± SD**	20.2 ± 3.8	22.2 ± 3.6	<0.001 *
**Residence status; living at home, *n* (%)**	82 (87.2)	1357 (97.7)	<0.001 *
**Pre-injury ADLs; walking w/ or w/o a cane, *n* (%)**	76 (81.7)	1354 (96.4)	<0.001 *
**Presence of a pre-injury comorbidity**			
Cerebrovascular disease, *n* (%)	19 (20.0)	126 (9.2)	<0.001 *
Parkinson’s disease, *n* (%)	3 (3.2)	19 (1.4)	0.162
Hypertension, *n* (%)	58 (61.1)	673 (48.8)	0.021 *
Diabetes mellitus, *n* (%)	29 (30.5)	301 (22.1)	0.057
Cardiovascular disease, *n* (%)	22 (23.2)	205 (15.1)	0.035 *
Respiratory disease, *n* (%)	11 (11.7)	70 (5.1)	0.007 *
Renal disease, *n* (%)	9 (9.5)	64 (4.7)	0.040 *
Musculoskeletal disorders, *n* (%)	21 (22.3)	169 (12.4)	0.006 *
**Number of medications, mean ± SD**	5.9 ± 4.6	3.8 ± 3.7	<0.001 *
**Modified 5-item frailty index, mean ± SD**	1.18 ± 0.86	0.77 ± 0.74	<0.001 *
**Blood test data at admission**			
Serum total protein value (g/dL), mean ± SD	6.6 ± 0.64	6.6 ± 0.71	0.762
Serum hemoglobin value (g/dL), mean ± SD	11.7 ± 1.9	12.8 ± 1.9	<0.001 *

ADLs, activities of daily living; SD, standard deviation; * *p* < 0.05.

**Table 2 jcm-12-01867-t002:** Comparison of the injury and treatment status between the dementia and non-dementia groups.

	Dementia (*n* = 95)	Non-Dementia (*n* = 1417)	*p*-Value
**Injury status**			
**Ground-level fall, *n* (%)**	54 (56.8)	525(37.1)	<0.001 *
**Cervical spine fracture, *n* (%)**	67 (70.5)	767 (54.1)	0.002 *
**Cervical dislocation, *n* (%)**	9 (9.5)	219 (15.5)	0.115
**Neurological impairment, *n* (%)**			0.056
AIS A or B, *n* (%)	10 (10.5)	193 (13.6)	
AIS C or D, *n* (%)	46 (48.4)	807 (57.0)	
No spinal cord injury, *n* (%)	39 (41.1)	417 (29.4)	
**Associated injuries, *n* (%)**	30 (31.6)	381 (26.9)	0.320
Head injury, *n* (%)	19 (20.0)	198 (14.0)	0.107
Thoracic injury, *n* (%)	5 (5.3)	83 (5.9)	0.819
Abdominal injury, *n* (%)	1 (1.1)	21 (1.5)	0.592
Upper limb injury, *n* (%)	4 (4.2)	73 (5.2)	0.459
Lower limb injury, *n* (%)	6 (6.3)	49 (3.5)	0.127
Pelvic fracture, *n* (%)	1 (1.1)	26 (1.8)	0.484
Thoracolumbar vertebral fracture, *n* (%)	7 (7.4)	84 (5.9)	0.571
**Treatment status**			
** ** **Steroid administration, *n* (%)**	9 (9.5)	198 (14.0)	0.216
** ** **Surgical treatment, *n* (%)**	47 (49.5)	856 (60.4)	0.035 *

AIS, American Spinal Injury Association Impairment Scale; * *p* < 0.05.

**Table 3 jcm-12-01867-t003:** Comparison of the adjusted variables between the propensity score-matched groups.

	Dementia (*n* = 61)	Non-Dementia (*n* = 61)	*p*-Value
**Age at injury (years), mean ± SD**	79.7 ± 6.4	79.4 ± 6.5	0.790
**Sex: Female, *n* (%)**	23 (37.7)	22 (36.1)	0.851
**Pre-injury** **ADLs** **: walking w/ or w/o a cane, *n* (%)**	58 (95.1)	59 (96.7)	0.500
**Neurological impairment**			0.405
AIS A or B, *n* (%)	6 (9.8)	8 (13.1)	
AIS C or D, *n* (%)	30 (49.2)	35 (57.4)	
No spinal cord injury, *n* (%)	25 (41.0)	18 (29.5)	
**Surgical treatment**	36 (59.0)	34 (55.7)	0.714

SD, standard deviation; ADLs, activities of daily living; AIS, American Spinal Injury Association Impairment Scale.

**Table 4 jcm-12-01867-t004:** Propensity score matching analysis of complications and ADLs after injury.

	Dementia (*n* = 61)	Non-Dementia (*n* = 61)	*p*-Value
**Complications, *n* (%)**	38 (62.3)	29 (47.5)	0.102
Respiratory impairment, *n* (%)	5 (8.2)	3 (4.9)	0.359
Dysphagia, *n* (%)	8 (13.1)	2 (3.3)	0.048 *
Pneumonia, *n* (%)	12 (19.7)	5 (8.2)	0.067
Urinary tract infection, *n* (%)	8 (13.1)	4 (6.6)	0.224
Deep venous thrombosis, *n* (%)	1 (1.6)	1 (1.6)	0.752
Pulmonary embolism, *n* (%)	2 (3.3)	0 (0)	0.248
Cerebral infarction, *n* (%)	0 (0)	2 (3.3)	0.248
Myocardial infarction, *n* (%)	0 (0)	0 (0)	N/A
Delirium, *n* (%)	16 (26.2)	9 (14.8)	0.116
**ADLs** **at 6 months: walking w/ or w/o a cane, *n* (%)**	28 (46.2)	42 (69)	0.016 *

ADLs, activities of daily living; AIS, American Spinal Injury Association Impairment Scale; N/A, not applicable; * *p* < 0.05.

## Data Availability

The datasets used and analyzed during the current study are available from the corresponding author on reasonable request.

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
