# Peer review of "Effects of Dementia on Outcomes after Cervical Spine Injuries in Elderly Patients: Evaluation of 1512 Cases in a Nationwide Multicenter Study in Japan"

_jcm, 2023, doi:10.3390/jcm12051867_

Round 1

Reviewer 1 Report

The manuscript elaborates on complications and increased risk and morbidity following cervical injury in patients with dementia. Cervical spine injuries in people with dementia can lead to serious consequences, including increased risk of morbidity and mortality, decreased quality of life, and difficulty recovering from surgery. There are many treatment options for cervical spine injuries in people with dementia, but these options can be difficult to use because of the patient's cognitive impairment and other medical conditions.

It is important to note that these studies are based on a retrospective design.
 There are some restrictions on these studies, such as the possibility of bias and data inaccuracies. Additionally, follow-up time may be constrained, and short-term results are not always comparable to long-term results.

1) Authors have mentioned dementia as a risk factor for low bone density. Please elaborate on the possible mechanisms where dementia could be a risk factor for low bone density compared to other factors that already exists in geriatric population. 

2) Similarly, presence of severe frailty in patients with dementia has been touted as possible reason for poor prognosis following cervical spine injury. Authors cite other papers to establish the association of frailty and dementia. 

3) I understand that this is a retrospective study. However, the manuscript would greatly benefit if the authors could put forward various possible biological and physiological mechanisms between the associations they mention or cite. 

4) There is evidence that people with dementia are at an increased risk of getting cervical spine injuries. This could be likely because they are more likely to fall. A segment of discussion about the incidence/ prevalence or risks of cervical injury in patients with dementia would be appropriate. 

Reviewer 2 Report

1. Is there any reason for select the 1,512 patients and what are the exclusion and inclusion criteria in selection?

2. Please double check the numbers in tables (p-values and dementia and non-dementia patients)

3. Need to change the figure 1, looks like Y axis (survival probability) numbers can modify to 0.0, 0.5 instead of 0.0 to 1.0.

4. I made some more comments in the manuscript
